# Shock Damage Analysis in Serial Femtosecond Crystallography Data Collected at MHz X-ray Free-Electron Lasers

**Alexander Gorel** [1,*,†], **Marie Luise Grünbein** [1,*,†], **Richard Bean** [2,‡], **Johan Bielecki** [2,‡], **Mario Hilpert** [1,*], **Michele Cascella** [2], **Jacques-Philippe Colletier** [3], **Hans Fangohr** [2,4,5], **Lutz Foucar** [1], **Elisabeth Hartmann** [1], **Mark S. Hunter** [6], **Henry Kirkwood** [2], **Marco Kloos** [1], **Romain Letrun** [2], **Thomas Michelat** [2], **Robert L. Shoeman** [1], **Jolanta Sztuk-Dambietz** [2], **Guillaume Tetreau** [3], **Herbert Zimmermann** [1], **Adrian P. Mancuso** [2,7], **Thomas R.M. Barends** [1,*], **R. Bruce Doak** [1], **Claudiu Andrei Stan** [8] **and Ilme Schlichting** [1]

[1]   Max Planck Institute for Medical Research, Jahnstrasse 29, 69120 Heidelberg, Germany;
      Lutz.Foucar@mpimf-heidelberg.mpg.de (L.F.); Elisabeth.Hartmann@mpimf-heidelberg.mpg.de (E.H.);
      Marco.Kloos@mpimf-heidelberg.mpg.de (M.K.); Robert.Shoeman@mpimf-heidelberg.mpg.de (R.L.S.);
      Herbert.Zimmermann@mpimf-heidelberg.mpg.de (H.Z.); bruce.doak@mpimf-heidelberg.mpg.de (R.B.D.);
      Ilme.Schlichting@mpimf-heidelberg.mpg.de (I.S.)
[2]   European XFEL GmbH, Holzkoppel 4, 22869 Schenefeld, Germany; richard.bean@xfel.eu (R.B.);
      johan.bielecki@xfel.eu (J.B.); michele.cascella@xfel.eu (M.C.); hans.fangohr@xfel.eu (H.F.);
      henry.kirkwood@xfel.eu (H.K.); romain.letrun@xfel.eu (R.L.); thomas.michelat@xfel.eu (T.M.);
      jolanta.sztuk@xfel.eu (J.S.-D.); adrian.mancuso@xfel.eu (A.P.M.)
[3]   Institut de Biologie Structurale, Université Grenoble Alpes, CEA, CNRS, 38044 Grenoble, France;
      colletier@ibs.fr (J.-P.C.); guillaume.tetreau@ibs.fr (G.T.)
[4]   Max Planck Institute for Structure and Dynamics of Matter, Luruper Chaussee 149,
      22761 Hamburg, Germany
[5]   University of Southampton, Southampton SO17 1BJ, UK
[6]   SLAC National Accelerator Laboratory, Menlo Park, CA 94025, USA; mhunter2@slac.stanford.edu
[7]   Department of Chemistry and Physics, La Trobe Institute for Molecular Science, La Trobe University,
      Melbourne, Victoria 3086, Australia
[8]   Department of Physics, Rutgers University Newark, 101 Warren Street, Newark, NJ 07102, USA;
      claudiu.stan@rutgers.edu
**\***   Correspondence: alexander.gorel@mr.mpg.de (A.G.); marie.gruenbein@mr.mpg.de (M.L.G.);
      mario.hilpert@mr.mpg.de (M.H.); thomas.barends@mr.mpg.de (T.R.M.B.)
†   Contributed equally.
‡   Contributed equally.

**Abstract:** Serial femtosecond crystallography (SFX) data were recorded at the European X-ray free-electron laser facility (EuXFEL) with protein microcrystals delivered via a microscopic liquid jet. An XFEL beam striking such a jet may launch supersonic shock waves up the jet, compromising the oncoming sample. To investigate this efficiently, we employed a novel XFEL pulse pattern to nominally expose the sample to between zero and four shock waves before being probed. Analyzing hit rate, indexing rate, and resolution for diffraction data recorded at MHz pulse rates, we found no evidence of damage. Notably, however, this conclusion could only be drawn after careful identification and assimilation of numerous interrelated experimental factors, which we describe in detail. Failure to do so would have led to an erroneous conclusion. Femtosecond photography of the sample-carrying jet revealed critically different jet behavior from that of all homogeneous liquid jets studied to date in this manner.

**Keywords:** X-ray free-electron laser; serial femtosecond crystallography; shock wave; protein crystallography

## 1. Introduction

By providing femtosecond pulses of extremely high peak brilliance, X-ray free-electron lasers (XFELs) enable novel experiments in structural biology. Biological microcrystals are often delivered to an XFEL beam in a liquid microjet [1], both to preserve biological integrity of jet contents and to provide rapid sample replenishment. The latter is mandatory, since each XFEL pulse annihilates the exposed sample. This holds true as well for a sample-delivery jet. An explosive vaporization not only opens a physical gap in the jet but can launch supersonic shock waves traveling up the jet into the oncoming sample [2]. This raises the concern, particularly at the 4.5 MHz maximum pulse rate of the European X-ray free-electron laser facility (EuXFEL) [2,3], that the accompanying pressure spike might affect the oncoming sample. The jet gap already sets limits on required jet speeds and permissible XFEL pulse rates requiring sufficient time between pulses for the continuous jet to have regenerated in the interaction region before arrival of the next XFEL pulse [4–6]. Shockwave damage would further exacerbate the limitations.

The EuXFEL operates in burst-mode, delivering XFEL pulses at up to 4.5 MHz within a 600 μs pulse train, with trains repeated at 10 Hz [7]. The long quiescent interval (0.1 s) between pulse trains ensures that the first pulse in each train probes undisturbed sample, suiting this mode particularly well to shock damage studies. The first pulse launches a shock wave traveling up the jet (and likewise the following pulses). By judicious choice of jet speed and XFEL pulse rate, the subsequent XFEL pulses will then probe samples exposed to one or more shock waves [2]. Several such damage assessments [4–6] have been carried out at the EuXFEL, employing MHz pulse rates and employing serial femtosecond crystallography (SFX) to record X-ray diffraction of protein microcrystals. Scattering from the first pulse in a train has been compared with that of subsequent pulses [4–6]. To date, under the XFEL beam conditions employed, no indication of damage has been found.

Here, we describe SFX experiments that we performed on lysozyme and myoglobin microcrystals at the SPB/SFX instrument [8] of EuXFEL at a maximum repetition rate of 1.129 MHz. To accelerate acquisition of "shock-free data" using the first pulses of a MHz bunch pattern, we exploited a newly available bunch patterning capability [9] and subdivided each X-ray train into 9 "wagons" separated by ~13–14 μs (Figure 1), a separation long enough to avoid shock damage (see the Results Section). Each wagon consisted of four consecutive pulses separated by 0.9 μs (~1.1 MHz repetition rate) followed by a fifth pulse after 1.8 μs (~0.55 MHz) (Figure 1). One can assess from this pattern whether the shock effect is additive (comparing data from second, third, and fourth pulses), reversible (fourth vs. fifth pulse), and perhaps even extract information on the shock damping time (comparing all pulses).

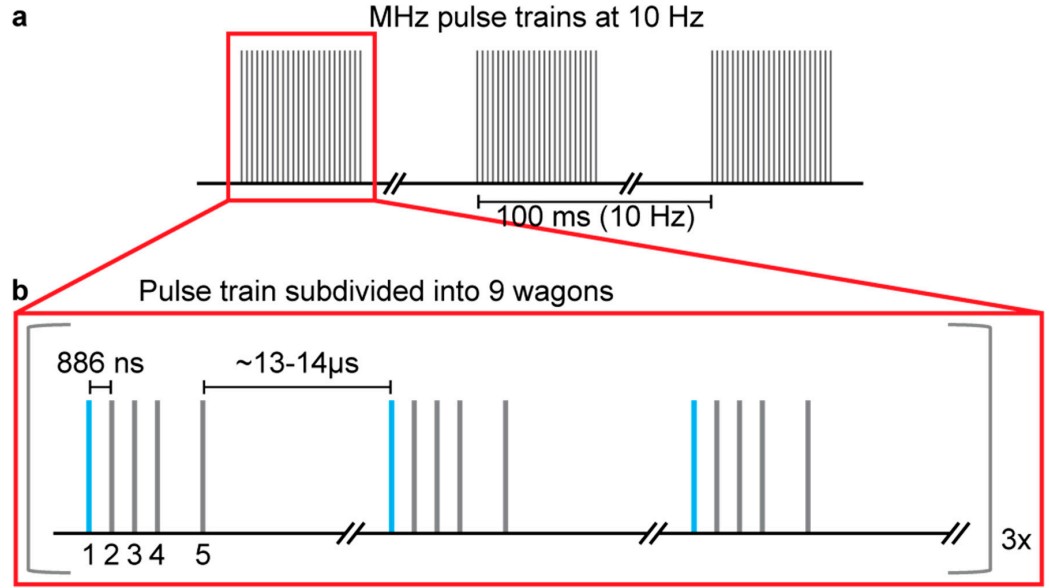

**Figure 1.** Pulse scheme at the European XFEL (EuXFEL). (**a**) The European XFEL delivers 10 bursts ("trains") of X-ray pulses per second. Each train can contain up to 2700 X-ray pulses, giving a maximum intra-burst pulse repetition rate of 4.514 MHz. (**b**) The novel pulse pattern employed in these measurements subdivided each pulse "train" into nine "wagons", separated by 13–14 μs, an inter-wagon spacing that allows any effects on the jet from the previous wagon to either convect away or damp out. The wagon structure comprised four pulses equally spaced by 0.886 μs (1.129 MHz effective pulse rate), followed by a fifth pulse at twice this spacing (1.772 μs or 0.564 MHz). This pattern allows to test whether shock wave-induced damage occurs (comparison of data collected using the first and later pulses, respectively), is additive (second vs. third or fourth pulse data), and reversible (last pulse vs. second to last pulse). Equally important, the wagon structure increases the rate of collection of damage-free data by a factor equal to the number of wagons per train (from the 10 Hz EuXFEL base rate up to 90 Hz, in our measurements).

## 2. Materials and Methods

### 2.1. Sample Preparation, Injection, and Jet Imaging

Lysozyme [10] and myoglobin [11] microcrystals were grown as described previously [10,11]. The injection and jet imaging were set up as described in previous experiments [2,4,12]. Briefly, the microcrystalline slurry was filtered using a 20 μm stainless-steel inline filter (IDEX Health & Science, IL, USA). Sample was injected via a liquid microjet from a gas dynamic virtual nozzle (GDVN) injector [1] using helium as the focusing gas. The sample concentration was adjusted to contain 10–15% (*v/v*) settled crystalline material. During injection, the sample was kept at 20 °C in a rotating temperature-controlled reservoir to prevent crystal settling [13]. The sample flow rate was 30–70 μL min$^{-1}$, and gas pressure was 400–620 psi at the inlet of the GDVN's gas supply line, corresponding to a flow rate of 140–250 mL min$^{-1}$. Sharp microscopic optical images of the jet were recorded once per pulse train using by optical flash photography using an EuXFEL femtosecond laser system [14,15] for illumination [2,4]. This illumination pulse and the camera itself were triggered by the EuXFEL 10 Hz global trigger, indicating the arrival of an X-ray pulse train. Images were recorded after a chosen delay time, set with nanosecond precision relative to the pulse train so as to image the jet shortly after a second X-ray pulse generated a second visible gap in the jet. By imaging two gaps in the jet, the jet speed could be determined from a single image. The measured jet speed of ~48 ± 5 m s$^{-1}$ was sufficient to transport a fresh sample segment into the interaction region between pulses. Moreover, online viewing of the images allowed fine-tuning of the sample flow rate and GDVN gas flow to ensure constant jet velocity even after the sample batch was changed.

## 2.2. Data Collection

Data collection was performed similarly as in previous experiments [4,12]. The experiment was performed at the SPB/SFX instrument [8] of EuXFEL (Proposal 2156, March 2019) with the accelerator producing ten pulse trains per second with a base intra-train rate of 1.129 MHz. During this experiment, up to 176 pulses/train were available to the SPB/SFX instrument. For a continuous train of pulses, only the first pulse in the train probes a jet segment guaranteed to be unaffected by previous pulses. We augmented data acquisition efficiency of guaranteed undamaged data by making use of bunch patterning. Each X-ray train was subdivided into 9 X-ray pulse bursts ("wagons") running at MHz intra-burst repetition rate (Figure 1). 14–15 pulses between wagons were dropped such that the last pulse of each wagon interacted with the jet 13–14 µs before the arrival of the first X-ray pulse of the subsequent wagon (Figure 1).

The photon energy was tuned to 9.3 keV. The X-ray focus was ~3 µm, and the electron bunch length was ~50 fs full width at half maximum (FWHM). For each individual X-ray pulse, the pulse energy was recorded by one of EuXFEL's X-ray gas monitor detectors (XGMD) [16] located upstream of the experimental hutch and was ~0.5–1.3 mJ on average. The SFX data of lysozyme and myoglobin microcrystals were collected in different shifts. An additional myoglobin dataset was collected in which each wagon consisted of three consecutive pulses separated by ~0.886 µs (~1.129 MHz repetition rate), followed by a fourth pulse after 2.658 µs (~0.376 MHz repetition rate). The position of the sample jet was continuously adjusted to maximize the hit rate.

## 2.3. Data Analysis for Number of Prior Pulses Launching Shocks

Knowing how often and how well the jet was intersected by XFEL pulses is essential, since it determines the number and magnitude of shocks experienced from previous pulses as well as the effective intensity with which the jet has been probed. Due to instabilities of the jet and to possible variations of the lateral pointing of the X-ray beam, a significant fraction of shots missed the jet partially or entirely and thus did not launch shock waves. For an analysis of potential shock effects on upstream sample quality, it therefore must be established for each pulse whether the jet was hit and how many of the preceding pulses of that wagon have also hit the jet. This can be done by analyzing the scattered intensity of the solvent in the recorded diffraction images. For fast and efficient characterization of scattered intensity in the solvent ring, we used data from a single detector module (module 02) that overlapped well with the solvent ring signal. To quantify solvent scattering intensity, the standard deviation $\sigma$ of all raw (non-calibrated) pixel intensities was employed. This proved to be a more robust measure than other variables, such as the median pixel intensity. The distribution of pixel intensities in the module becomes bimodal when solvent scattering is present. Hence, the higher the solvent scattering intensity, the larger the standard deviation in pixel intensities, making $\sigma$ an excellent proxy for intensity of solvent scattering. It was empirically found that solvent scattering is visible on the detector for $\sigma \gtrsim 400$, which was subsequently used as a threshold to identify whether the jet was hit.

To identify the number of shocks $N$ launched by previous pulses, the solvent scattering strength $\sigma$ was stored together with the trainID $t$ and cellID $c$ of each shot that uniquely identify it and its relative position to other shots. For each shot $i$ hitting the jet (i.e., $\sigma_i \geq 400$) with trainID $t_i$ and cellID $c_i$, the solvent scattering strength $\sigma_{i-1}$ of the previous shot (trainID $t_i$, cellID $c_{i-1}$) was interrogated. If the jet was hit ($\sigma_{i-1} \geq 400$), the number of pre-shocks $N_i$ for shot $i$ was increased by 1 and shot $i-2$ was analyzed. If this again hit the jet, $N_i$ was again increased by 1, and shot $i-3$ was analyzed. This iterative procedure was terminated either when the beginning of an X-ray wagon was reached or at the first occurrence of not hitting the jet.

## 2.4. Data Processing

Data processing was performed based on an approach developed in previous experiments [4,12]. Data from the Adaptive Gain Integrating Pixel detector (AGIPD) [17] was calibrated using the

calibration pipeline established at the EuXFEL [18], with constants provided by the facility and the AGIPD consortium. The CFEL-ASG Software Suite (CASS) [19] version 2.4.0 was used for online data analysis and offline hit identification and fed by a ZeroMQ [20,21] data stream from EuXFEL [22]. A hit is defined as an image in which more than ten Bragg spots were identified. To this end, we used the algorithm described in Reference [19]. Indexing and integration were performed with CrystFEL version 0.8.0 [23]. The sample-detector distance was nominally 120.2 mm. The positions and orientations of individual sensor modules of the AGIPD were refined as described elsewhere [11]. The resolution of a diffraction pattern was determined by that of the outermost (highest scattering angle) indexed Bragg reflection, with a ratio of its intensity I and $\sigma(I)$ exceeding a given threshold (signal to noise ratio, SNR, $I/\sigma(I)$). Many diffraction images of the myoglobin microcrystals contain intense salt diffraction peaks (SNR > 20, resolution higher than 1.6 Å) due to deposition of salt (from the sample buffer) on the injector tip. Therefore, for these datasets, we excluded resolution regions higher than 1.6 Å before analyzing the resolution of the Bragg peaks from protein crystals. For determining resolution at a given SNR of hits in the myoglobin dataset, the outermost crystal reflection within a given signal-to-noise ratio range (e.g., $3 \leq SNR \leq 3.5$) has to be analyzed to exclude salt peaks impairing analysis.

Due to slightly different accelerator behavior between different data collection shifts, the pulse energy distribution over a given pulse train differs for the two datasets. The range of variation was less during the acquisition of myoglobin than for lysozyme data, but the pulse energy still changed by up to 40% over the train and by almost 10% within wagons. The significant variations in pulse energy can largely be attributed to the novelty of this mode of accelerator operation at the time of the experiment (March 2019), and it prevented us from merging the data collected across all wagons within a given pulse train. Since then, the pulse energy stabilization has reportedly been greatly improved, so such merging should be feasible in future experiments.

## 3. Results

The SFX datasets described here were collected from lysozyme and myoglobin microcrystals at the SPB/SFX instrument [8] of EuXFEL, utilizing XFEL pulse rates of up to 1.129 MHz. We used bunch patterning [9] to effectively collect data at 90 Hz rather than 10 Hz, the train repetition rate. Each X-ray pulse train was divided into 9 "wagons" separated by ~13–14 μs. This time interval is significantly longer than the 4–5 μs needed to replace the ~200 μm free-standing jet segment destroyed by the last XFEL pulse of the preceding wagon (from the nozzle to the interaction region) flowing at 40–50 m s$^{-1}$. Thus, between wagons, the full length of the jet was replenished along with any possibly damaged crystals. Further upstream of the free-standing jet, in the GDVN nozzle meniscus region, the jet diameter increases from the ~5 μm jet diameter to the 75 μm diameter of the sample capillary. As a shock traverses this meniscus, energy conservation ensures that the associated pressure jump diminishes rapidly with penetration into the meniscus. Moreover, the pressure jump has already decreased significantly even before the shock reaches the meniscus, given that the pressure jump damps exponentially with travel distance within the jet, as shown by Blaj et al. [3]. Consequently, samples within the meniscus at the time of shock wave passage could be considered damage-free. By this criterion, two consecutive X-ray pulses separated by 13–14 μs ensured data collection of samples undamaged by shock waves. Each wagon consisted of four consecutive pulses separated by 0.886 μs (~1.129 MHz repetition rate), followed by a fifth pulse after 1.772 μs (~0.564 MHz) (Figure 1). In total, 10,087,200 diffraction images were collected, 3,766,500 of the lysozyme and 6,320,700 myoglobin crystals, respectively. Of those images, 491,120 (13.0%) were hits for lysozyme and 425,819 (6.7%) for myoglobin. The final number of indexed diffraction patterns is 315,157 (64.2%) for lysozyme and 194,034 (45.6%) for myoglobin, with the resolution limit of the Monte-Carlo integrated data being 1.8 Å in both cases. We deposited the images containing hits of lysozyme and myoglobin microcrystals in the Coherent X-ray Imaging Data Bank website (CXIDB) [24] with the CXIDB ID 144 at (http://cxidb.org/id-144.html). In addition to the deposited diffraction data, we provide experiment metadata, such as the pulse energy and images of the sample jet for further analysis.

### 3.1. Analysis of all Crystal Hits

As in previous work [4,6], we first merged and sorted SFX patterns according to their position within X-ray wagons, then compared "shock-free" and "shocked" datasets. With increasing number of previous pulses, and thus increasing shock exposure from previous pulses, the resolution appeared unchanged (indicating no damage, Figure 2a), yet the indexing rate appeared to decrease (indicating damage, Figure 2b). This quandary prompted a thorough and detailed examination of all metadata, which revealed numerous intertwining issues.

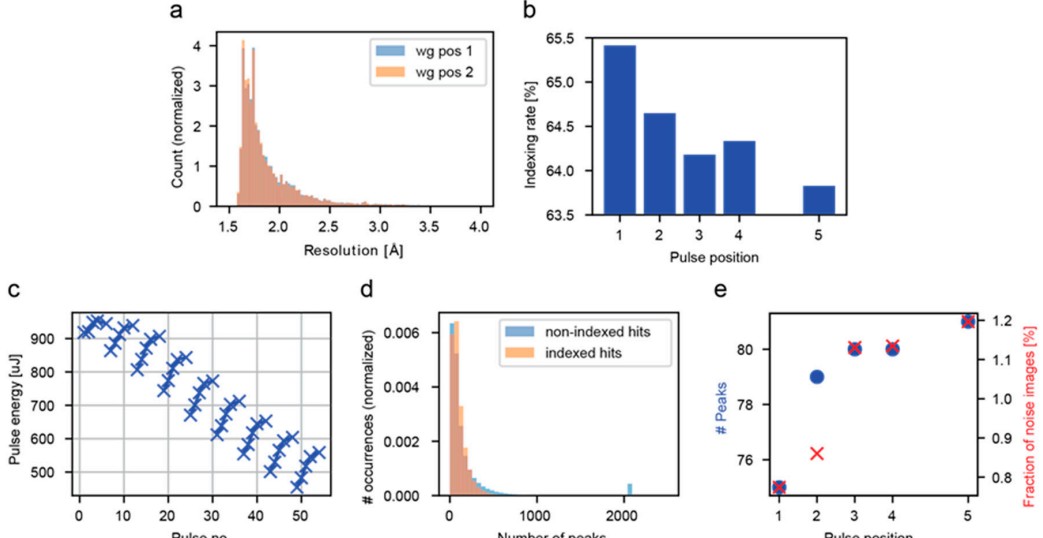

**Figure 2.** Comparison of data quality: first vs. subsequent pulses in a wagon. (**a**) Histogram of the resolution (at signal-to-noise ratio (SNR) 3) of diffraction data collected using pulses 1 and 2 of all X-ray wagons. (**b**) Indexing rate as a function of pulse position averaged over all wagons. (**c**) Average pulse energy for all pulses in the X-ray trains. (**d**) Histogram of the number of peaks detected in diffraction patterns that can (orange) and cannot (blue) be indexed. None of the indexed diffraction patterns contain more than 1140 peaks. (**e**) Average number of peaks (blue circles) and fraction of hits containing more than 1140 peaks (red crosses) as a function of pulse position in a wagon, averaged over all wagons. (**a**–**e**) show results from diffraction data of lysozyme microcrystals.

Experimental properties (pulse energy, detector behavior, etc.) were found to vary systematically pulse-by-pulse within wagons, as well as wagon-by-wagon within trains. The higher pulse energies of the last pulses in each wagon (Figure 2c) ensues an increase in the mean intensity of a diffraction image, which in turn leads to an increased number of diffraction peaks (Figure 2e). However, concomitantly, the fraction of images containing a peculiarly large number of diffraction peaks increases, which translates into an increase of the number of diffraction images with substantial noise (Figure 2d,e), lowering the indexing rate (Figure 2b). The apparent decrease of the indexing rate along the wagon is therefore an artefact generated by intertwining of these dependencies.

### 3.2. Analysis of Hits in Continuous Jets

When analyzing the femtosecond images of the jet, we noted dramatic large-scale motions of the jet (Figure 3). The issue of jet instabilities has been known for a long time, but it was believed to be happening on a relatively long timescale. When observing the jet during data collection, it seemed to "jump" from time to time, requiring adjustments to ensure jet/X-ray overlap. Unexpectedly, on top of these slow motions, very fast jet movements also occur, as evidenced by our femtosecond jet imaging. This means that one cannot assume that for any given shot the jet was also intersected by the previous X-ray pulse. Hence, it is not clear whether a given crystal actually experienced a shock wave launched by a previous pulse, since this requires a continuous liquid jet column between the shots. Therefore,

a rigorous shock investigation first requires an analysis of whether or not a continuous jet existed between subsequent XFEL exposures. We examined this by analyzing the intensity of the water ring in the diffraction images.

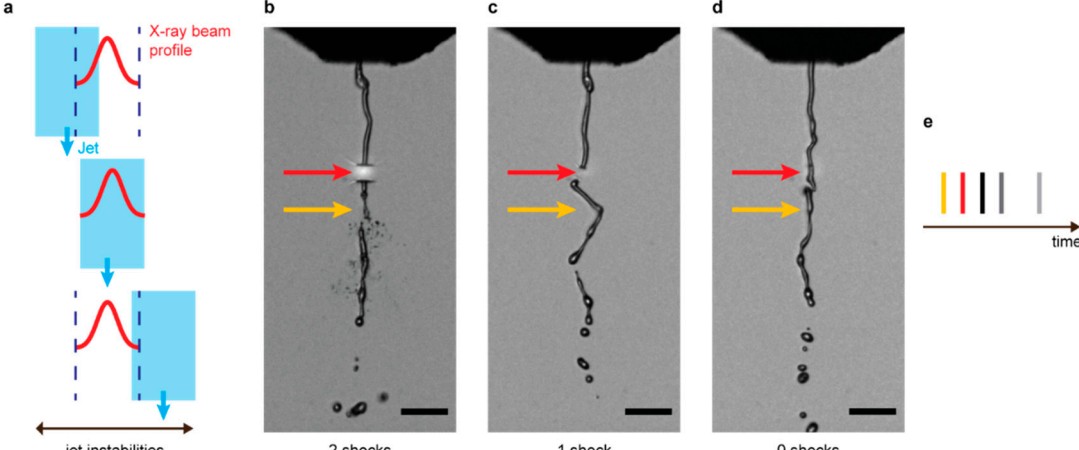

**Figure 3.** Jet stability can influence the observed effect of previous pulses on data quality. (**a**–**d**) The liquid microjet carrying the microcrystals can show significant deviations from the well-behaved straight geometry shown in (**b**). Sideways wiggling of the jet (**c**,**d**), or similarly, variations in XFEL pointing, can cause the XFEL beam to partially or entirely miss the jet (**a**). Jet instabilities are more pronounced with relatively large microcrystals or high crystal densities and affect not only the apparent hit rate but also the number and strength of shock waves launched by one pulse to affect later ones. For example, data collected with the third pulse in a train/wagon can have experienced two (**b**), one (**c**) or zero (**d**) shocks depending on the detailed motion of the jet as the individual pulses arrive. The yellow and red arrows indicate the positions along the jet where the first and second pulse of the wagon are expected to hit. Whether a given X-ray pulse hits the jet and launches a shock wave can be examined by analyzing the intensity of the water ring in the diffraction images. (**b**) Two gaps at the expected interaction regions (on the jet) of the first and second pulse are clearly visible in the continuous jet. Both pulses most likely intersected the jet and launched shock waves propagating upstream. (**c**) There is no gap visible at the jet position where the first pulse intersects the jet (yellow arrow), it missed the jet due to wiggling. However, there is a gap visible at the location where the second X-ray pulse is expected to interact with the jet (red arrow). Thus, sample probed by the third pulse was subjected to one shock wave. (**d**) Wiggling of the jet (in the plane perpendicular to the X-ray beam axis) precludes interaction of either the first or the second pulse with the jet. No shockwave is launched. (**e**) Pulse pattern within a wagon.

We observed an increase in solvent scattering intensity with pulse energy, as expected. Moreover, we detected a correlation between the pulse energy at a given wagon position and the number of previous pulses also hitting the jet (Figure 4a). This is indicative of a better overlap/alignment of X-rays and jet, possibly due to higher beam stability. The increase in pulse energy in turn leads to an improvement in the mean resolution as a function of the number of shocks experienced (Figure 4b). To compensate for this effect, we selected from the indexed data only the cases where all five X-ray pulses in the wagon hit the jet, meaning that only data of comparable jet/X-ray alignment quality were selected and within this subset, we compared data only within wagons, shown exemplary for wagon 1 in Figure 4c,d. The latter excludes effects from the dramatic change in pulse energy over the whole pulse train (Figure 2c). For these cases, the indexing rate (Figure 4c) and mean crystal diffraction resolution (Figure 4d) show no dependence on pulse position. After carefully incorporating these additional constraints, and only then, could we conclude that the data gave no evidence of shock-induced damage under our operating conditions. The corresponding data statistics are shown in Table 1.

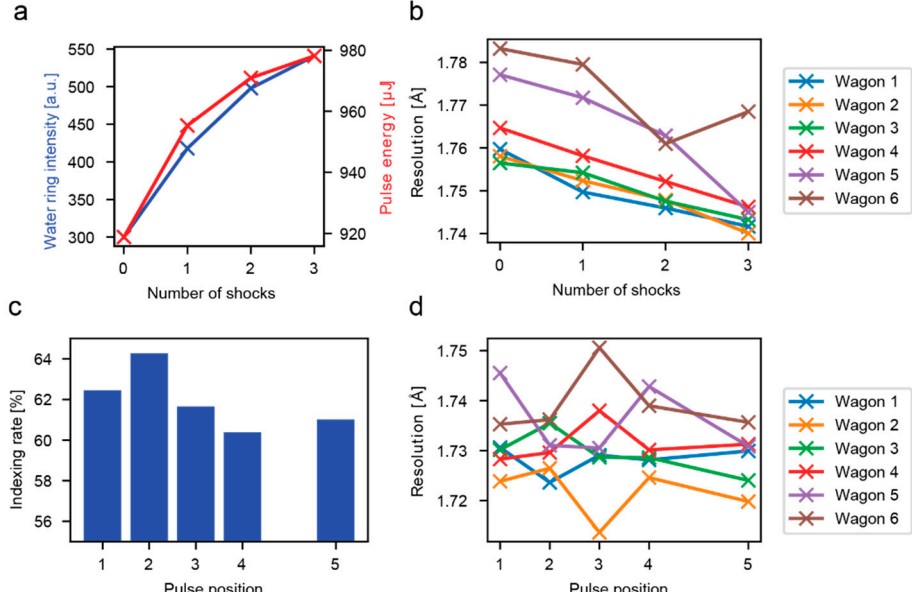

**Figure 4.** Comparison of data quality in continuous jets: first vs. subsequent pulses in a wagon. (**a**) Solvent scattering intensity and pulse energy at wagon position 4 correlate with the number of shock waves launched by previous pulses. A possible explanation could be that the X-ray pointing stability within a wagon increases with higher pulse energy, leading to improved alignment between X-rays and jet. (**b**) Diffraction resolution (at SNR 3) of indexed diffraction patterns at wagon position 4 as a function of the number of shock waves launched by previous pulses. The improvement in resolution is due to the concomitant increase in pulse energy (**a**). (**c**) Indexing rate as a function of pulse position for pulses in wagon 1, containing only data from wagons in which all 5 consecutive pulses had hit the jet. (**d**) Diffraction resolution (at SNR 3) as a function of pulse position for pulses in wagons 1–6, containing only data from wagons in which all 5 consecutive pulses had hit the jet. (**a**–**d**) Results from diffraction data of lysozyme microcrystals—the results for myoglobin microcrystals are very similar.

**Table 1.** Data collection statistics. All indexed images were selected where the X-rays hit the sample jet consecutively, i.e., in all five pulses of a wagon. The diffraction data of the first wagon for lysozyme and the sixth wagon for myoglobin were chosen because their pulse energies were most similar between shots. The resolution was set to 1.8 Å and used to calculate the data quality statistics. $CC_{1/2}$ describes the correlation of one half of the dataset with the other half of the dataset and is used to select the high-resolution cutoff for data processing [25].

| Lysozyme Run 20–52 | First Pulse of First Wagon | Second Pulse of First Wagon | Third Pulse of First Wagon | Fourth Pulse of First Wagon | Fifth Pulse of First Wagon |
|---|---|---|---|---|---|
| Space group | $P4_32_12$ | $P4_32_12$ | $P4_32_12$ | $P4_32_12$ | $P4_32_12$ |
| Cell dimensions (Å, °) | 79. 79. 39. 90. 90. 90. | 79. 79. 39. 90. 90. 90. | 79. 79. 39. 90. 90. 90. | 79. 79. 39. 90. 90. 90. | 79. 79. 39. 90. 90. 90. |
| # indexed images | 843 | 896 | 831 | 808 | 881 |
| Resolution (Å) | 22.83–1.80 (1.85–1.80) | 26.36–1.80 (1.85–1.80) | 26.37–1.80 (1.85–1.80) | 21.95–1.80 (1.85–1.80) | 22.83–1.80 (1.85–1.80) |
| $I/\sigma(I)$ | 1.6 (1.0) | 2.0 (1.0) | 1.5 (0.6) | 1.5 (1.2) | 1.5 (0.8) |
| $R_{split}$ (%) | 60.0 (126.7) | 60.2 (112.8) | 60.9 (115.8) | 61.2 (116.5) | 58.7 (129.6) |
| $CC_{1/2}$ | 0.604 (0.215) | 0.61 (0.296) | 0.585 (0.333) | 0.575 (0.265) | 0.608 (0.177) |
| CC* | 0.868 (0.595) | 0.87 (0.676) | 0.859 (0.707) | 0.854 (0.648) | 0.87 (0.548) |
| Completeness (%) | 99.4 (98.6) | 99.4 (99.1) | 99.3 (98.8) | 99.4 (98.8) | 99.5 (99.7) |
| Multiplicity | 11.9 (7.7) | 12.1 (7.9) | 11.9 (8.1) | 11.6 (7.6) | 12.8 (8.4) |
| Wilson B ($Å^2$) | 14.9 | 14.9 | 15.3 | 15.0 | 15.1 |
| Correlation on Is with entire wagon (%) | 91.8 | 92.1 | 91.9 | 91.6 | 92.6 |

**Table 1.** *Cont.*

| Myoglobin Run 66–107 | First Pulse of Sixth Wagon | Second Pulse of Sixth Wagon | Third Pulse of Sixth Wagon | Fourth Pulse of Sixth Wagon | Fifth Pulse of Sixth Wagon |
|---|---|---|---|---|---|
| Space group | $P2_1$ | $P2_1$ | $P2_1$ | $P2_1$ | $P2_1$ |
| Cell dimensions (Å,°) | 63.6 28.8 35.6 90. 106.5 90. | 63.6 28.8 35.6 90. 106.5 90. | 63.6 28.8 35.6 90. 106.5 90. | 63.6 28.8 35.6 90. 106.5 90. | 63.6 28.8 35.6 90. 106.5 90. |
| # indexed images | 971 | 823 | 858 | 881 | 917 |
| Resolution (Å) | 21.26–1.80 (1.85–1.80) | 22.34–1.80 (1.85–1.80) | 21.26–1.80 (1.85–1.80) | 22.34–1.80 (1.85–1.80) | 19.85–1.80 (1.85–1.80) |
| $I/\sigma(I)$ | 2.1 (1.3) | 2.0 (1.6) | 2.0 (1.8) | 2.0 (1.4) | 2.1 (1.7) |
| $R_{split}$ (%) | 64.9 (95.0) | 66.4 (99.6) | 63.3 (92.4) | 64.0 (94.2) | 66.8 (90.3) |
| $CC_{1/2}$ | 0.42 (0.102) | 0.417 (0.245) | 0.456 (0.534) | 0.427 (0.154) | 0.42 (0.251) |
| CC* | 0.769 (0.429) | 0.767 (0.628) | 0.792 (0.834) | 0.774 (0.517) | 0.769 (0.634) |
| Completeness (%) | 97.9 (93.3) | 96.6 (94.8) | 97.1 (94.4) | 96.9 (94.4) | 97.7 (95.5) |
| Multiplicity | 10.6 (7.0) | 8.8 (5.8) | 9.5 (6.3) | 9.2 (6.1) | 10.2 (6.4) |
| Wilson B (Å$^2$) | 12.9 | 13.7 | 14.5 | 14.7 | 13.1 |
| Correlation on Is with entire wagon (%) | 91.2 | 89.6 | 90.2 | 90.6 | 91.1 |

The quality of diffraction data of protein crystals, including of SFX data [4,6,26], is often judged by the strength of the anomalous signal of sulfur atoms. To this end, it is common to refine the protein structure and show an anomalous electron density map around methionine residues or disulfide bridges [4,6,26]. Because of the extensive data filtering and selection, there are <1000 indexed images per pulse position, which is fewer than is typically used for structure determination. However, the values of the established data quality indicators of the diffraction intensities (Table 1) are within the range expected, particularly in view of the relatively low number of merged images, which would not allow the Monte Carlo integration to fully converge. Moreover, the intensities of each of the individual datasets show a high correlation (~90%) with the intensities derived from the images from the entire wagon, which does contain sufficient images for Monte Carlo convergence.

## 4. Discussion

Our SFX experiments were performed with lysozyme and myoglobin microcrystals at the SPB/SFX end station [8] of the EuXFEL, utilizing XFEL pulse rates of up to 1.129 MHz using pulse patterning [9]. When we merged and sorted SFX patterns according to equivalent pulse positions in the wagon, then compared "shock-free" (first pulse in a wagon) and "shocked" (all others) sets, we observed that, with increasing wagon position (nominal shock exposure), the resolution remained invariant (indicating no damage), yet the indexing rate decreased (indicating damage). To unravel this conundrum, a thorough examination of the entire dataset was undertaken. Numerous intertwining factors were discovered (Figure 2). Experimental properties (pulse energy, detector behavior, etc.) were found to vary systematically pulse-by-pulse within wagons as well as wagon-by-wagon within trains. The femtosecond images revealed dramatic large-scale motion of the jet (Figure 3) on the sub-microsecond timescale, highly questioning the implicit assumption in previous analysis of shock damage analysis [4–6] that one can simply average diffraction patterns collected at a certain position along the pulse train to obtain a shocked dataset. The fast jet movements may also be a reason for relatively low hit rates observed in some SFX experiments so far.

Data analysis became a matter of culling a sub-set of data conforming to identical experimental attributes. We compared data only for pulses of comparable pulse energy within a given wagon and used the integrated scattering intensity of the solvent ring to indicate, for each probe event, exactly how many shock waves had been launched by previous pulses within that wagon, taking this as the truly experienced shock exposure. To maximize damage signal and to ensure that a linear section of jet had been exposed and probed, we considered only situations of comparable X-ray/jet alignment leading to

maximum shock exposure to each wagon (all pulses in a wagon hit the jet). With this reduced, coherent data sub-set, we found no evidence of shock wave damage. Delving into the interlocked complexities of XFEL beam, jet, and detector was critical to this deduction. A less comprehensive analysis would have supported an erroneous conclusion.

Due to instabilities in the accelerator operation—which are now resolved—we could not make use of the advantages of the new pulse pattern: It not only speeds up data collection by a factor corresponding to the number of wagons in a train, but also allows to assess whether the shock effect is additive (by comparing data accumulated from second, third, and fourth pulses), reversible (fourth vs. fifth pulse), and possibly even extract the shock damping time (by comparing all pulses). Future experiments aiming at addressing the question of shock damage at 4.5 MHz at EuXFEL will greatly benefit from the lessons learned and systematic approaches described here.

**Author Contributions:** M.L.G., M.K., R.L.S. and R.B.D. performed injection; E.H., H.Z. and I.S. prepared samples; M.H. and L.F. performed online data analysis using CASS; A.G., J.-P.C., M.L.G. and T.R.M.B. performed offline analysis; R.B., J.B., M.S.H., R.L., H.K. and A.P.M. set up the SPB/SFX Instrument and collected the data; J.S.-D. and M.C. operated the AGIPD detector; H.F. and T.M. performed online data analysis and development; J.B., R.L., M.L.G. and C.A.S. set up femtosecond jet imaging; I.S. and C.A.S. conceived and designed the experiment; M.L.G., A.G., R.B., J.B., M.S.H., A.P.M., R.L., H.K., J.S.-D., G.T., M.C., H.F., T.M., J.-P.C., E.H., H.Z., R.B.D, M.K., R.L.S., M.H., L.F., T.R.M.B., C.A.S. and I.S. prepared the experiment; M.L.G., A.G., R.B.D. and I.S. wrote the paper with input from all authors. All authors have read and agreed to the published version of the manuscript.

**Funding:** This research was supported by the Max Planck Society and travel grants from the European XFEL. C.A.S. was supported by startup funds from Rutgers University-Newark. Contributions by J.-P.C. and G.T. were supported by the Agence Nationale de la Recherche (grants ANR-17-CE11-0018-01 to J.-P.C.).

**Acknowledgments:** We acknowledge European XFEL in Schenefeld, Germany, for provision of X-ray free electron laser beamtime (proposal 2156) at the SPB/SFX instrument and thank the instrument group and facility staff for their assistance. We thank Dirk Noelle for very helpful discussions on beam and pulse train properties. We thank the AGIPD consortium for providing some of the calibration constants for the detector. The authors are indebted to the SFX User Consortium for the provision of instrumentation and personnel that have enabled this experiment. We are grateful to Filipe Maia for very helpful discussions and uploading the data to CXIDB.ORG.

**Conflicts of Interest:** The authors declare no conflict of interest.

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
