# Peer review of "Shock Damage Analysis in Serial Femtosecond Crystallography Data Collected at MHz X-ray Free-Electron Lasers"

_crystals, doi:10.3390/cryst10121145_

Round 1

Reviewer 1 Report

This study investigated the influence of supersonic shock wave with microjets of protein microcrystals with the irradiation of XFEL pulse. By using the different pulse train in XFEL, serial femtosecond crystallography (SFX) data with the different patterns of X-ray irradiation are measured and are confirmed that there is no evident influence of the XFEL irradiation. These fundamental data are important for SFX experiments. I think it is suitable to publish Crystals as the present form of the manuscript.

Author Response

We thank the reviewer for the positive feedback on our work.

Reviewer 2 Report

The manuscript by Gorel et al. describes their results of the experiments on “shock damage analysis in SFX irradiated by the MHz X-FEL beam”. The experiment is well designed, however, because of various instabilities of sample injection, X-FEL pulses etc., conclusions form the results are not clear. Further experiments with more stable X-FEL beam and improved sample injection will be necessary to make a clear conclusion. I agree with the authors that the experiments described in this manuscript was good lessens for the future experiments. From these reasons, I cannot judge whether the manuscript will be accepted or not.

Minor point:

Figure 3 shows quite interesting results, but it is difficult to understand why the third pulse in a train/wagon experienced 2, 1 or 0 shocks. I would like to ask the authors to improve the caption.

Author Response

We agree with the reviewer that the approach to experiment and analysis described in our manuscript is important for further experiments at EuXFEL investigating the potential effects of shock wave induced damage. We thank the referee for pointing out that Figure 3 requires a more explicit explanation to improve comprehensibility to the reader and have done so in the revised version of our manuscript.
